# High-quality reference genome of *Fasciola gigantica*: Insights into the genomic signatures of transposon-mediated evolution and specific parasitic adaption in tropical regions

**Xier Luo[1,2☯], Kuiqing Cui[2☯], Zhiqiang Wang [2☯], Zhipeng Li[2], Zhengjiao Wu[2], Weiyi Huang[2], Xing-Quan Zhu[3], Jue Ruan[1,2]\*, Weiyu Zhang[2]\*, Qingyou Liu [1,2☯]\***

**1** Shenzhen Branch, Guangdong Laboratory of Lingnan Modern Agriculture, Genome Analysis Laboratory of the Ministry of Agriculture and Rural Affairs, Agricultural Genomics Institute at Shenzhen, Chinese Academy of Agricultural Sciences, Shenzhen, China, **2** State Key Laboratory for Conservation and Utilization of Subtropical Agro-bioresources, Guangxi University, Nanning, China, **3** College of Veterinary Medicine, Shanxi Agricultural University, Taigu, China

☯ These authors contributed equally to this work.

\* ruanjue@caas.cn (JR); zweiyu@gxu.edu.cn (WZ); qyliu-gene@gxu.edu.cn (QL)

**Data Availability Statement:** The whole genome assembly (contig version) and gene annotation reported in this paper have been deposited in the

## Abstract

*Fasciola gigantica* and *Fasciola hepatica* are causative pathogens of *fascioliasis*, with the widest latitudinal, longitudinal, and altitudinal distribution; however, among parasites, they have the largest sequenced genomes, hindering genomic research. In the present study, we used various sequencing and assembly technologies to generate a new high-quality *Fasciola gigantica* reference genome. We improved the integration of gene structure prediction, and identified two independent transposable element expansion events contributing to (1) the speciation between *Fasciola* and *Fasciolopsis* during the Cretaceous-Paleogene boundary mass extinction, and (2) the habitat switch to the liver during the Paleocene-Eocene Thermal Maximum, accompanied by gene length increment. Long interspersed element (LINE) duplication contributed to the second transposon-mediated alteration, showing an obvious trend of insertion into gene regions, regardless of strong purifying effect. Gene ontology analysis of genes with long LINE insertions identified membrane-associated and vesicle secretion process proteins, further implicating the functional alteration of the gene network. We identified 852 predicted excretory/secretory proteins and 3300 protein-protein interactions between *Fasciola gigantica* and its host. Among them, copper/zinc superoxide dismutase genes, with specific gene copy number variations, might play a central role in the phase I detoxification process. Analysis of 559 single-copy orthologs suggested that *Fasciola gigantica* and *Fasciola hepatica* diverged at 11.8 Ma near the Middle and Late Miocene Epoch boundary. We identified 98 rapidly evolving gene families, including actin and aquaporin, which might explain the large body size and the parasitic adaptive character resulting in these liver flukes becoming epidemic in tropical and subtropical regions.

Genome Warehouse in BIG Data Center, Beijing Institute of Genomics (China National Center for Bioinformation), Chinese Academy of Sciences, under accession number GWHAZTT00000000 that is publicly accessible at https://ngdc.cncb.ac.cn/gwh. The AGP file for Hi-C was uploaded as supplement file. The Pacbio sequencing reads has been deposited into the genome sequence archive (GSA) in BIG under accession code CRA003783. The whole genome assembly also can be obtained in the National Center for Biotechnology Information (NCBI) under Bioproject PRJNA691688.

**Funding:** This work was supported by the National Natural Science Fund (grants No. U20A2051, and 31860638) to QL, National Natural Science Fund (grants No. 31760648) to KC, National Natural Science Fund (grants No. 31960706) to WZ, Guangxi Natural Science Foundation (grants No. AB18221120) to QL, Guangxi Distinguished Scholars Program (grants No. 201835) to QL, and Science and Technology Major Project of Guangxi (grants No. Guike AA17204057) to WZ. The funders had no role in study design, data collection and analysis, decision to publish, or preparation of the manuscript.

**Competing interests:** The authors have declared that no competing interests exist.

## Author summary

Fascioliasis is a neglected zoonotic tropical disease of humans, which reduces the productivity of animal industries, and imposes an economic burden of at least 3.2 billion dollars annually. Although there are four assemblies for *F. hepatica* and two assemblies for *F. gigantica* at NCBI, the inherent limited ability of short reads based assemblies made the completeness of genome sequences and the quality of gene annotation challenging. Here, we report the Pacbio assembly of reference genome for *F. gigantica*, and the quality of assembly and gene annotation are significantly improved compared with previous assemblies. Besides, we found the evidence of transposon-mediated evolution, especially for LINE insertions into gene regions between 41 Ma and 62 Ma, contributing to the speciation and adaption of the *Fasciola* ancestors. Furthermore, we identified *F. gigantica* specific gene duplication including 98 gene families, and 3300 protein-protein interactions between *F. gigantica* and the host in the small intestine and liver environment. These results illustrate the genomic and gene evolution of *F. gigantica* potentially shaping multiple parasitic characters.

## Introduction

*Fasciola gigantica* and *Fasciola hepatica*, known as liver flukes, are two species in the genus *Fasciola*, which cause fascioliasis commonly in domestic and wild ruminants, but also are causal agents of fascioliasis in humans. Fascioliasis reduces the productivity of animal industries, imposes an economic burden of at least 3.2 billion dollars annually worldwide, and is a neglected zoonotic tropical disease of humans, according the World Health Organization's list [1]. *F. gigantica*, the major fluke infecting ruminants in Asia and Africa, has been a serious threat to the farming of domesticated animals, such as cows and buffaloes, and dramatically reduces their feed conversion efficiency and reproduction [2]. The prevalence of *F. gigantica* infection has greatly affected subsistence farmers, who have limited resources to treat their herds, and has hindered economic development and health levels, especially in developing countries.

The various omics technologies provide powerful tools to advance our understanding of the molecules that act at the host-parasite interface, and allow the identification of new therapeutic targets against fascioliasis [3]. To date, four assemblies for *F. hepatica* and two assemblies for *F. gigantica* have been deposited at the NCBI [4–7]. These assemblies reveal a large genome with a high percentage of repeat regions in *Fasciola* species, and provided valuable insights into features of adaptation and evolution. However, these assemblies are based on the short read Illumina sequencing or hybrid sequencing methods, with limited ability to span large families of repeats. Various limitations have led to the current assemblies in the genus *Fasciola* being fragmented (8 kb to 33 kb and 128 kb to 1.9 Mb for contig and scaffold N50s, respectively). Subsequent gene annotation analysis using current assemblies were also challenging, with abundant transposition events occurring over evolutionary history, which significantly increased the repeat components in intron regions, resulting in considerable fragmentation in gene annotation.

Infection by *Fasciola* causes extensive damage to the liver, and excretory/secretory (E/S) proteins play an important role in host-parasite interactions. Parasite-derived molecules interact with proteins from the host cell to generate a protein interaction network, and these proteins partly contribute to *Fasciola*'s striking ability to avoid and modulate the host's immune response [8]. Previous proteomics of E/S proteins have highlighted the importance of secreted

extracellular vesicles (EVs) and detoxification enzymes to modulate host immunity by internalizing with host immune cells [9,10]. The anthelminthic drug, triclabendazole (*TCBZ*), is currently the major drug available to treat fascioliasis at the early and adult stages, which acts by disrupting β-tubulin polymerization [11]; however, over-reliance on *TCBZ* to treat domesticated ruminants has resulted in selection for resistance to liver flukes [12]. Drug and vaccine targets for molecules associated with reactive oxygen species (ROS)-mediated apoptosis have recently been validated as an effective tools in multiple helminth parasites [13]. Increased understanding of host-parasite and drug-parasite interactions would facilitate the development of novel strategies to control fascioliasis.

In recent years, there have been increasing numbers of human cases of fascioliasis, becoming a major public health concern in many regions [14,15]. However, high quality genome assemblies for liver flukes are still insufficient. In the present study, we combined multiple sequencing technologies to assemble a chromosome-level genome for *F. gigantica* and provided integrated gene annotation. Protein-protein interactions were analyzed between the predicted *F. gigantica* secretome and host proteins expressed in the small intestine and liver. In addition, gene family analysis identified a series of genes expansions in *F. gigantica*. Interestingly, the distribution of repeat sequences in the genome exhibit an excess of long interspersed element (LINE) duplications inserted into intronic regions, potentially helping to explain the duplications of transposable element (TE) plasticizing gene structures and possibly acting as long-term agents in the speciation of *Fasciola*.

## Results

### Pacbio long reads-based *de novo* assembly and gene annotation

The *F. gigantica* genome contains abundant repeat sequences that are difficult to span using short read assembly methods, and the complex regions also hinder integrated gene annotation of the genome. Therefore, in the present study, multiple sequencing technologies, have been applied: (1) Single-molecule sequencing long reads (~91× depth) using the Pacbio Sequel II platform; (2) paired-end reads (~66× depth) using the Illumina platform; and (3) chromosome conformation capture sequencing (Hi-C) data (~100× depth) (S1 Table). The initial assembly was performed using the Pacbio long reads, followed by mapping using single-molecule sequencing and Illumina sequencing reads to polish assembly errors and sequencing mistakes, resulting in a contig N50 size of 4.89 Mb (Fig 1A). The Hi-C data were used to build final super-scaffolds, resulting in a total length of 1.35 Gb with a scaffold N50 size of 133 Mb (Fig 1B and S1 Fig and Table 1 and S1–S3 Tables). The final assembly consists of 10 pseudo-chromosomes covering more than 99.9% of the *F. gigantica* genome, and the length distribution was approximate equal to the estimation by karyotype in previous research (S2 Fig and S4 Table) [16]. The assessment of nucleotide accuracy shows that the error rate was $5.7 \times 10^{-6}$ in the genome. QUAST analysis [17] showed a high mapping and coverage rate using both Illumina short reads and Pacbio long reads, in which 99.73% of reads mapped to 99.85% of the genome with more than 10× depth (S5 Table).

Combing *de novo*/homolog/RNA-seq prediction, a total of 12,503 protein coding genes were annotated in the *F. gigantica* genome. BUSCO assessment [18] indicated that the genome is 90.4% complete and 5.6% fragmented, underscoring the significant improvement of the genome continuity and gene-structure predictions compared with previous assemblies (S6 Table). Specifically, the average gene length in the annotated data is 28.8 kb, nearly twice the length of that in other digenean species, but contrasted with the similar average length of the coding sequences (CDSs). Through functional annotation, we found that 8569 of the genes could be characterized in the InterPro database [19,20], 7892 of them were mapped to the gene

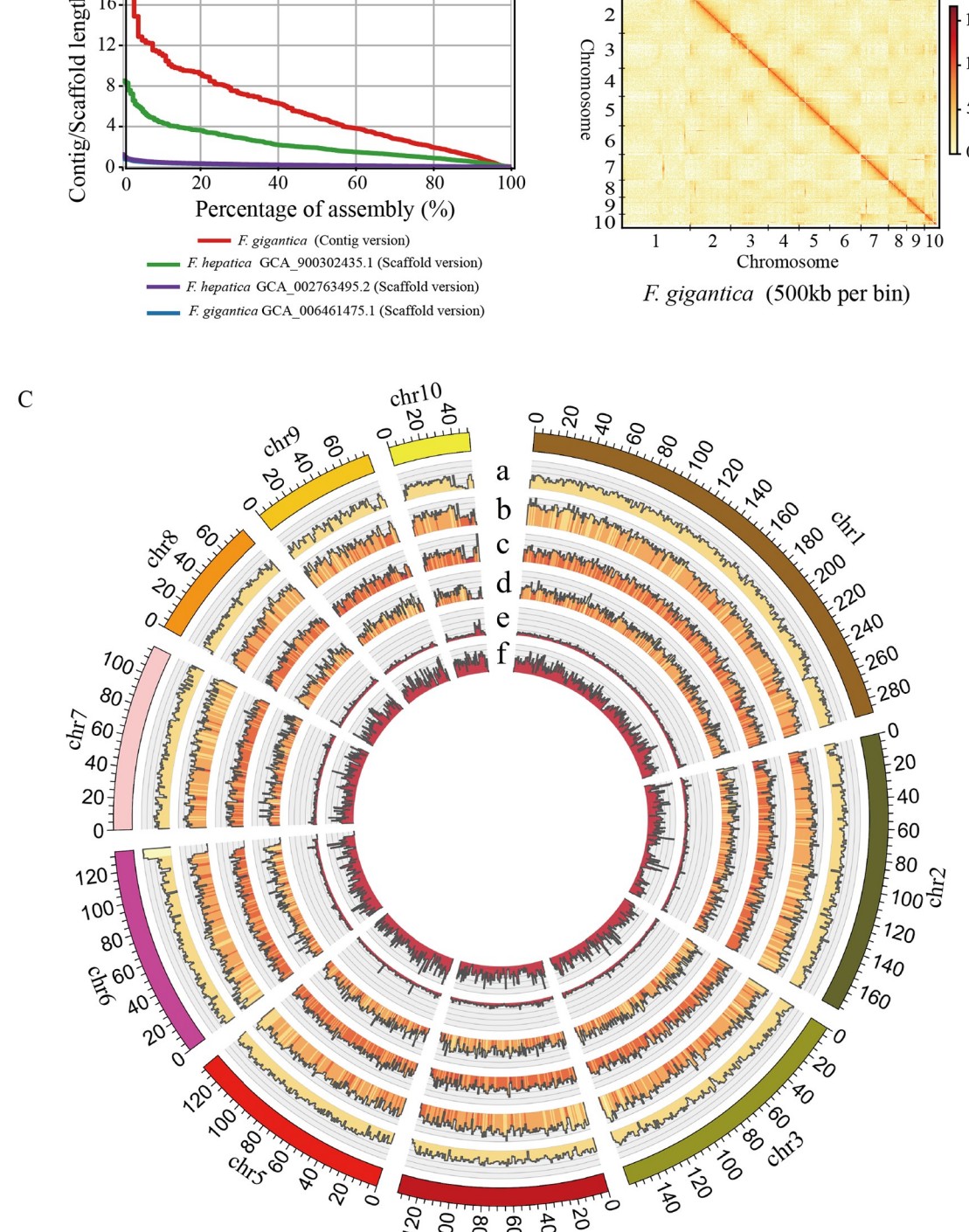

**Fig 1. Landscape of the *Fasciola gigantica* genome.** (A) Comparisons of the assembled contigs and scaffold lengths (y-axis) and tallies (x-axis) in *Fasciola* species. (B) Hi-C interactive heatmap of the genome-wide organization. The effective mapping read pairs between two bins were used as a signal of the strength of the interaction between the two bins. (C) Integration of genomic and annotation data using 1 Mb bins in 10 Hi-C assembled chromosomes. (a) Distribution of the GC content (GC content > 39% and < 52%); (b) distribution of the long interspersed element (LINE) percentage > 0% and < 50%; (c) distribution of the long

terminal repeat (LTR) percentage > 0% and < 50%; (d) distribution of the gene percentage > 0% and < 70%; (e) distribution of the heterozygosity density of our sample (percentage > 0% and < 1%); (f) distribution of the heterozygosity density of SAMN03459319 in the NCBI database. Hi-C, chromosome conformation capture sequencing;

ontology (GO) terms, and 5353 of them were identified by the Kyoto Encyclopedia of Genes and Genomes (KEGG) pathways database (S3–S4 Figs and S7 Table).

## The unique repeat duplications in *Fasciola*

TEs are insertional mutagens and major drivers of genome evolution in eukaryotes, and replication of these sequences, resulting in variation of gene structure and expression, have been extensively documented [21,22]. Besides, TEs are molecular fossils, being remnants of past mobilization waves that occurred millions of years ago [23]. In the present study, we identified repeat sequences combined the analysis from RepeatModeler [24] and RepeatMasker [25], and detected a significant proportion of them neglected by previous studies. In the *F. gigantica* genome, we identified 945 Mb of repeat sequences, which was approximate 20% more than that identified in other assemblies in *Fasciola* species, while the lengths of non-repeat sequences were nearly identical. The most convincing explanation for the additional assembled repeat sequences was that the contigs constructed from Pacbio long reads spanned longer repeat regions, which were compressed in previous assemblies. Among these repeat sequences, there were 408 Mb of LINEs (corresponding to 30.3% of the assembled genome), 285 Mb of long terminal repeats (LTRs, corresponding to 21.2% of the assembled genome), and 162 Mb of unclassified interspersed repeats (corresponding to 12.0% of the assembled genome) (S5 Fig and S8 Table). According to the repeat landscapes, we found that there were two shared expansion events for LINEs and LTRs that occurred approximately 12 million years ago (Ma) and 65 Ma, and an additional expansion event at 33 Ma for LTRs (S6–S7 Figs). Our result confirmed previous study on family Fasciolidae [6], and the abundant repeat sequences in the *Fasciola* genomes aroused the interest concerning the role of repeats in evolution (Fig 2A), which implied a hypothesize that the expansion of TEs enlarged the genome size of an ancestor of

**Table 1. Summary statistics for the genome sequences and annotation.**

| | | *F. gigantica* |
|---|---|---|
| Genome | Total Genome Size (Mb) | 1,348 |
| | Chromosome Number | 10 |
| | Scaffold Number [a] | 10+24 |
| | Scaffold N50 (Mb) | 133 |
| | Scaffold L50 | 4 |
| | Contig Number | 1,022 |
| | Contig N50 (Mb) | 4.89 |
| | Heterozygosity Rate (%) | $1.9 \times 10^{-3}$ |
| Annotation | Total Gene Number | 12,503 |
| | Average CDS Length (bp) | 1552.7 |
| | Average Gene Length (kb) | 28.8 |
| | Percentage of Genome Covered by CDSs (%) | 1.5% |
| | BUSCO Assessment | 90.4% |
| | Repeat Content | 70.0% |

[a] number of chromosome level scaffolds and unplaced scaffolds. CDS, coding sequence.

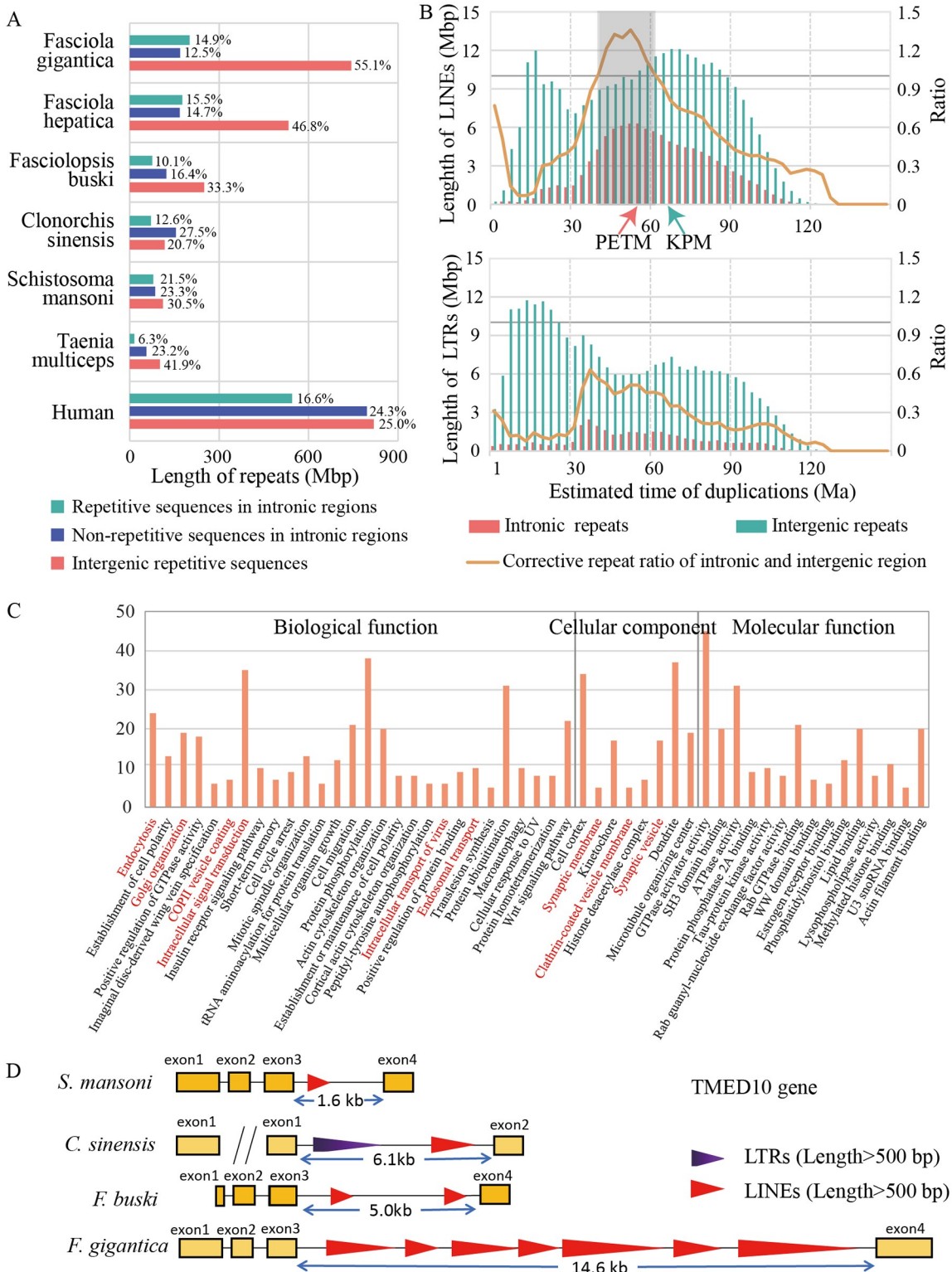

**Fig 2. Identification of repeat expansion and alternative gene networks in the *Fasciola gigantica* genome.** (A) The distribution of repetitive sequence length among the genomes of six flatworms and the human genome. (B) Landscape of LINEs and LTRs distribution in the *Fasciola gigantica* genome. The x-axis shows the expansion time of TEs calculated by the divergence between repeat sequences. The mutation rate was set as $1.73 \times 10^{-9}$ per year. The orange line represents the repeat length ratio, used to estimate the signatures of selection, which was corrected by the total length of intronic and intergenic regions in history. (C) The functional enrichment of genes

with more than 10 kb LINE insertions between 41 Ma and 62 Ma by Gene Ontology (GO) classification. The GO terms related to vesicle secretion are marked in red. (D) *TMED10* gene structure map. LINEs original between 41 Ma and 62 Ma and longer than 500 bp identified by RepeatMasker were plotted. LTRs longer than 500 bp were plotted. Long interspersed element, LINE; long terminal repeat, LTR; TE, transposable element; TMED10, transmembrane P24 trafficking protein 10.

*Fasciola* to gain a new advantage by rewiring gene networks. To test this hypothesis, we focused on the genome-wide repeats distribution and test signatures of selection.

For new TE insertions to persist through vertical inheritance, transposition events must be under strong purifying effect among gene loci to avoid disturbing their biological function. However, we observed many intronic repeat elements in *Fasciola*, resulting in a larger intron size per gene. If there are equal selection effects on newly inserted TEs in intronic and intergenic regions, there would be a high correlation between the distribution of insertion time and retained TE lengths between these two regions. By contrast, there would be fewer accumulated repeat sequences existing under purifying effect. In this study, we use the relative proportion of TEs between intronic and intergenic regions as a simple indicator, and use the inferred size of intronic and intergenic regions over evolutionary history as a control to estimate the signatures of selection. The results showed that TE insertions into intronic regions are under persistent intense purifying effect, except for LINEs. There was an excess of persistent LINE insertions into intronic regions between 41 Ma and 62 Ma, indicating different modes of accumulating LINEs into intronic regions compared with that in other periods (Fig 2B). Specifically, the time of the ancient intronic LINE expansion (~51.5 Ma) was different to the genome-wide LINE expansion time (~68.0 Ma), whereas the time was coincident with two important environmental change events, the Cretaceous-Paleogene boundary (KPB) mass extinction (~66.0 Ma) and the Paleocene-Eocene Thermal Maximum (PETM) (~55.8 Ma). Both the PETM and KPB events recorded extreme and rapid warming climate changes; however, rapid evolutionary diversification followed the PETM event, as opposed to near total mass extinction at the KPB [26]. Therefore, we selected genes with different LINE lengths, derived between 41 Ma and 62 Ma, and expected to identify a transposon-mediated alterative gene network contributing to the host switch and the shift from intestinal to hepatic habitats.

## LINE-mediated alterative gene network

We identified a substantial proportion of genes with LINE insertions, derived between 41 Ma and 62 Ma, indicating a universal effect of the gene network. We selected 1288 genes with the LINE insertions of more than 10 kb, representing more than one third of the average gene length, and annotated the genes using Gene Ontology (GO) terms and processes and Kyoto encyclopedia of genes and genomes (KEGG) pathways (Fig 2C and S9–S11 Tables). These genes involve molecules internalizing substances from their external environment, including membrane-associated and vesicle secretion process proteins. Meanwhile, the gene network was likely adapted to the evolution of protein biosynthesis and modification of histones.

Enrichment analysis of GO terms showed that membrane and membrane-associated proteins are over-represented, involving "synaptic membrane" ($P$ = 3.52E-04), "clathrin-coated vesicle membrane" ($P$ = 1.08E-03), and "synaptic vesicle" ($P$ = 3.02E-03), as well as vesicles secretion processes, such as "endocytosis" ($P$ = 7.06E-06), "Golgi organization" ($P$ = 7.45E-05), "COPII vesicle coating" ($P$ = 2.72E-04), "intracellular signal transduction" ($P$ = 5.16E-04), and "endosomal transport" ($P$ = 2.47E-03). Besides, proteins relating to phosphorylation and GTPase activators were also enriched, such as "Protein phosphorylation" (p = 1.73E-03), "Regulation of small GTPase mediated signal transduction" ($P$ = 1.21E-03), "GTPase activator activity" ($P$ = 1.13E-10). The over-representation of genes involved in membrane transport and

signal transduction was particularly interesting because helminth parasites interfere with the host immune system by secreting molecules from surface tegument or gut. The *TMED10* gene in *F. gigantica* (encoding transmembrane P24 trafficking protein 10) was used as an example. *TMED10* is a cargo receptor involved in protein vesicular trafficking along the secretory pathway [27,28], and the genes have an 11.1 kb LINE insertion in the third intron, resulting in an over three-fold increment in the gene length (Fig 2D). The enrichment suggests that the gene network related to secretion could have experienced adaptive evolution during LINE transposition events. We further compared our dataset with the proteome result from *F. hepatica* extracellular vesicles (EVs) [9], and found 21 proteins that were also identified as surface molecules associated with EV biogenesis and vesicle trafficking (*IST1*, *VPS4B*, *TSG101*, *MYOF*, *ATG2B*, *STXBP5L*, and 15 Rho GTPase-activating related proteins). Specifically, *IST1*, *VPS4B*, and *TSG101* are members of the endosomal sorting complex required for transport (ESCRT) pathway, which promotes the budding and release of EVs. *TSG101*, a crucial member of the ESCRT-I complex, has an important role in mediating the biogenesis of multi-vesicular bodies, cargo degradation, and recycling of membrane receptors. Besides, the ESCRT pathway promotes the formation of both exosomal carriers for immune communication. During the formation of the immunological synapse between T-cells and antigen-presenting B cells, *TSG101* ensures the ubiquitin-dependent sorting of T-Cell Receptor (*TCR*) molecules to exosomes that undergo *VPS4*-dependent release into the synaptic cleft[29].

The most significant KEGG pathway was aminoacyl-tRNA biosynthesis ($P$ = 7.16E-04), containing 15 out of 38 annotated aminoacyl tRNA synthetases (*AARSs*). *AARSs* are the enzymes that catalyze the aminoacylation reaction by covalently linking an amino acid to its cognate tRNA in the first step of protein translation. The large-scale insertion of LINEs reside in *AARS* genes suggested that the ancestor of *Fasciola* may have profited from the effect of transposition, with changes to protein biosynthesis and several metabolic pathways for cell viability. In addition, a significant number of genes are strongly associated with histone modulation, including "histone deacetylase complex" ($P$ = 1.89E-03), "histone methyltransferase activity (H3-K36 specific)" ($P$ = 1.08E-03), and "methylated histone binding" ($P$ = 2.37E-03). Histone modifications play fundamental roles in the manipulation and expression of DNA. We found nine histone deacetylases and Histone methyltransferases in the gene set (*HDAC4*, *HDAC8*, *HDAC10*, *KMT2E*, *KMT2H*, *KMT3A*, *KDM8*, *NSD1*, and *NSD3*). Histone modifications can exert their effects by influencing the overall structure of chromatin and modifying and regulating the binding of effector molecules [30,31]; therefore, the variation of these genes might bring about evolution from a disturbed gene structure to a mechanism of genome stabilization to tackle a continuous genome amplification process in evolutionary history.

## Genome-wide host-parasite interaction analysis

In the *Fasciola* genome, we predicted genes encoding 268 proteases, 36 protease inhibitors (PIs), and 852 predicted excretory/secretory (E/S) proteins that are commonly involved in interacting with hosts and modulating host immune responses (S8 Fig). The largest class of proteases was cysteine peptidases (n = 113), which was also identified in the *F. hepatica* genome (Fig 3A and S12 Table). The largest (n = 19, 52.8% of PIs) PI family was the I02 family of Kunitz-BPTI serine protease inhibitors, which bind to Cathepsin L with a possible immunoregulatory function [32] (S13 Table). GO enrichment analysis of E/S proteins showed that proteins related to "activation of cysteine-type endopeptidase activity" ($P$ = 6.14E-19), "peroxidase activity" ($P$ = 3.79E-07) and "protein disulfide isomerase activity" ($P$ = 3.75E-06) are over-represented (Fig 3B, S14–S15 Tables). Indeed, there were 38 cysteine peptidases identified as E/S

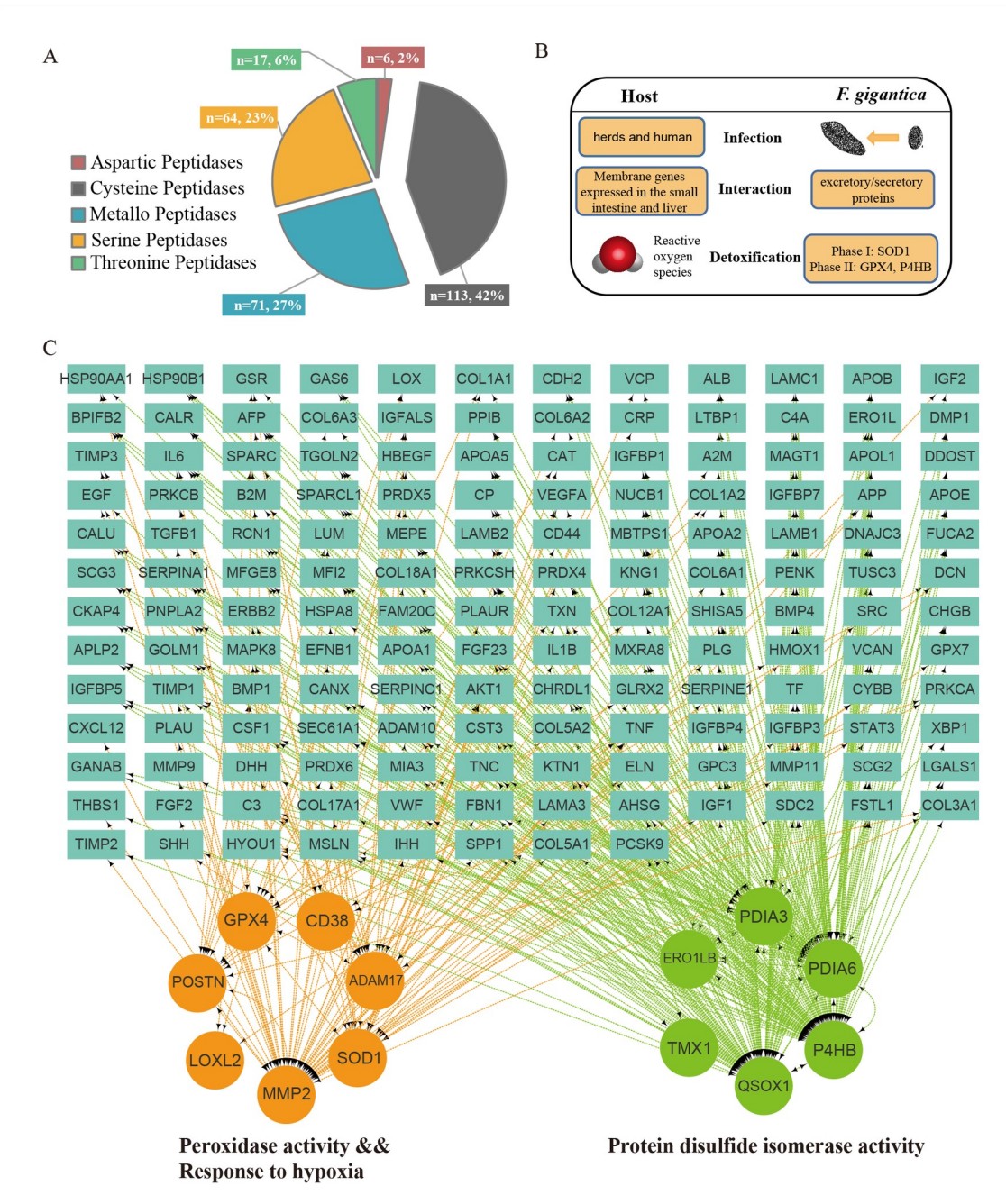

**Fig 3. Genome-wide host-parasite interaction analysis.** (A) Pie chart for proteases identified in *Fasciola gigantica*. (B) The interaction mode between the adult *Fasciola gigantica* and the host. (C) The protein-protein interaction (PPI) network of redox-related pathways in *Fasciola gigantica* with host proteins. The genes indicated in the three gene ontology (GO) terms were significantly enriched and have their encoded proteins have PPIs with excretory/secretory (E/S) proteins.

proteins, including cathepsin L-like, cathepsin B-like, and legumain proteins, which participate in excystment, migration through gut wall, and immune evasion [33].

In parasites, as in mammalian cells, ROS are produced as a by-product of cell metabolism and from the metabolism of certain pharmacological agents. The ability of a parasite to survive

in its host has been directly related to its antioxidant enzyme content [34]. To further analyze host-parasite interactions, we identified the protein-protein interactions (PPIs) between the *F. gigantica* secretome and human proteins expressed in the small intestine and liver [8]. In total, we identified 3300 PPIs, including rich interactions that directly or indirectly participated in the two phases of detoxification pathways (Fig 3C). Superoxide dismutase [Cu-Zn] (*SOD*, PPIs = 49) was first highlighted because of its important role on phase I detoxification against ROS, in which it catalyzes the dismutation of the superoxide radical to molecular oxygen and hydrogen peroxide ($H_2O_2$) [35]. Gene family analysis identified six *SOD* paralogs in *F. gigantica*, and two of them contained a signal peptide (Fig 4D). Previous enzyme activity assays also confirmed a significant difference between *SOD* activities and concentration in E/S proteins of two *Fasciola* species [36], suggesting an intense ability to resist superoxide radical toxicity. Meanwhile, the metabolite of phase I, $H_2O_2$, can also damage parasites, which requires detoxification enzymes, including glutathione-dependent enzymes *GPx*, glutathione reductase, and other peroxidases. Protein disulfide-isomerase (*P4HB*, PPIs = 132) and phospholipid hydroperoxide glutathione peroxidase (*GPX4*, PPIs = 28) were as functioning in phase II detoxification. GPx catalyzes the reduction of hydroperoxides (ROOH) to water, using glutathione (*GSH*) as the reductant. P4HB also participates in the process by mediating homeostasis of the antioxidant glutathione [37]. However, we did not identify E/S proteins in the Cytochrome P450 (*CYP450*) family in phase III detoxification. Therefore, we speculated that successful parasite defense against *F. gigantica* mainly depends on the strong superoxide activity and efficient hydrogen peroxide detoxification.

## Gene family analysis

Gene family analysis was performed using eight taxa (*F. gigantica*, *F. hepatica*, *Fasciolopsis buski*[38], *Clonorchis sinensis* [39], *Schistosoma mansoni*)[40], *Taenia multiceps* [41], swamp buffalo [42], and human [43], which identified 17,992 gene families (Fig 4A). Phylogeny analysis of 559 single-copy orthologs showed that *F. gigantica* and *F. hepatica* shared a common ancestor approximately 11.8 million years ago (2.2–22.5 Ma, 95% highest posterior density [HPD]) near the Middle and Late Miocene Epoch boundary. The Miocene warming began 21 million years ago and continued until 14 million years ago, when global temperatures took a sharp drop at the Middle Miocene Climate Transition (MMCT). The divergence of the two *Fasciola* species may have resulted from the consequences of rapid climate changes, such as migration of the host causing geographic isolation. Our estimation is between the previously suggested date of 5.3 Ma based on 30 nuclear protein-coding genes[6], and 19 Ma based on cathepsin L-like cysteine proteases [44]. Although we used a more integrative gene dataset, the wide HPD interval could not be neglected, raising possible uncertainty from the complex process of speciation or inappropriate protein sequence alignment between members of the genus *Fasciola*.

The distribution of gene family size among different species is used to estimate which lineages underwent significant contractions or expansions. Compared with *F. hepatica*, *F. gigantica* shows more gene family expansion events (643 compared to 449) and a similar number of gene family contractions (713 compared to 672). The result emphasizes the general trend that, relative to the common ancestor of *Fasciola*, the ancestor of *F. gigantica* apparently underwent a higher extent of gene-expansion than did the ancestor of *F. hepatica*. Gene duplication is one of the primary contributors to the acquisition of new functions and physiology [45]. We identified 98 gene families, including 629 genes, as rapidly evolving families specific to *F. gigantica*. Family analysis showed a fascinating trend of gene duplication, with substantial enrichment for the "structural constituent of cytoskeleton" ($P$ = 3.52E-24), "sarcomere organization"

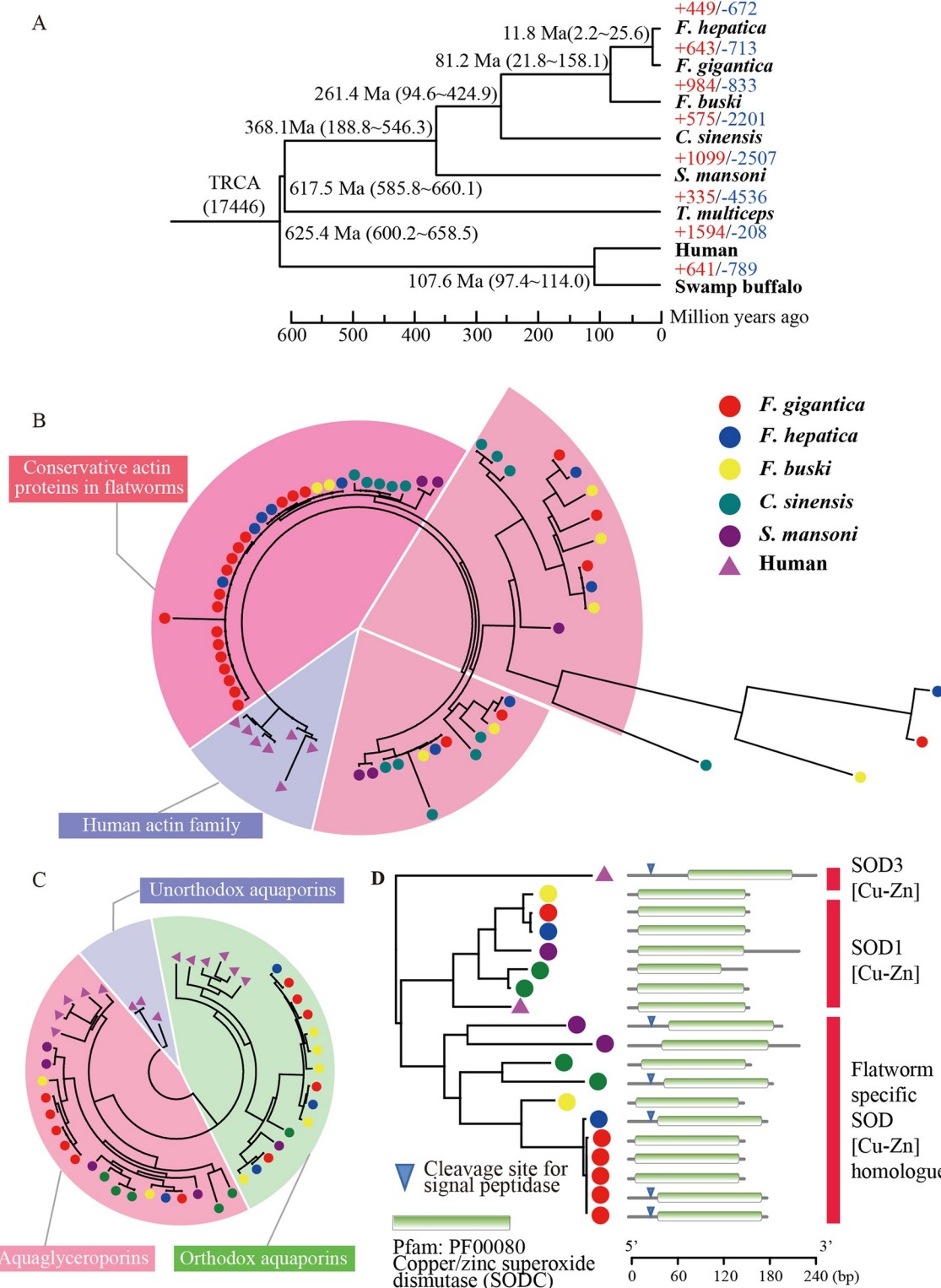

**Fig 4. Phylogenetic tree and gene family analysis.** (A) A phylogenetic tree generated using 559 single-copy orthologous genes. The numbers on the species names are the expanded (+) and contracted (-) gene families. The numbers on the nodes are the divergence time between species. (B) A phylogenetic tree of actin genes in flatworms and humans. All human homologue genes are selected as outgroup. (C) Phylogenetic tree of aquaglyceroporin (AQP) family genes in flatworms and humans. The human homolog genes (*AQP11*, *AQP12A*, and *AQP12B*) were selected as the outgroup. (D) A phylogenetic tree of copper/zinc superoxide dismutase (*SOD*) genes in flatworms and humans. The midpoint was selected as the root node.

($P$ = 2.29E-14), "actin filament capping" ($P$ = 6.19E-13), and "spectrin" ($P$ = 3.03E-11) in *F. gigantica* (S16 Table). There were 24 actin paralogs in *F. gigantica*, in contrast to 8 actin paralogs in *F. hepatica*. Actin is one of the most abundant proteins in most cells, and actin filaments, one of the three major cytoskeletal polymers, provide structure and support internal movements of organisms [46]. They are also highly conserved, varying by only a few amino acids between algae, amoeba, fungi, and animals [47]. We observed three types of actin proteins in flukes, according to their identity from human actin family. Seventeen of the 24 actin proteins in *F. gigantica* are highly conserved (Identity > 95%) (Fig 4B). Consistent with the accepted role of the epidermal actin cytoskeleton in embryonic elongation [48,49], we speculated that the significant expansion of actin and spectrin genes increased the body size of *F. gigantica* via cell elongation or proliferation during morphogenesis. Another rapidly evolving family is the aquaglyceroporin subfamily in the membrane water channel family. We found six aquaglyceroporin paralogs in *F. gigantica*, which were over-represented in the GO term "water transport" ($P$ = 2.10E-06) (Fig 4C). Aquaglyceroporins are highly permeated by glycerol and other solutes, and variably permeated by water, as functionally validated by several studies [50,51]. The mammalian aquaglyceroporins regulate glycerol content in epidermal, fat, and other tissues, and appear to be involved in skin hydration, cell proliferation, carcinogenesis, and fat metabolism. A previous study showed that *F. gigantica* could withstand a wider range of osmotic pressures compared with *F. hepatica* [52], and we speculated that a higher aquaglyceroporin gene copy number might help explain this observation.

It is worth mentioning that 57.6% of the rapidly evolving expansion genes specific to the *F. gigantica* genome were driven by tandem duplication, such that the newly formed duplicates preserved nearly identical sequences to the original genes. The newly formed genes would accumulate non-functionalizing mutations, or develop new functions over time. We found only few tandem duplicated genes that had non-functionalizing mutations, suggesting that adaptive evolution could have an important role in the consequences of these genes via a dosage effect or neo-functionalization.

## Discussion

The genome of *Fasciola* species contains a large percentage of repeat sequences, making them the largest parasite genomes sequenced to date. Since the first assembly of *F. hepatica* was submitted in 2015 [5], several studies have aimed to improve the quality of assembly and gene annotation [4,6,7]. With advances in long read sequencing assembly and Hi-C scaffolding technologies, it is now viable to resolve the genomic "dark matter" of repetitive sequences, and other complex structural regions at relatively low cost [53]. Therefore, we present the highest quality genome and gene annotation for *F. gigantica* to date, and provide long-awaited integrated genome annotation for fascioliasis research.

In previous study of Fasciolidae family, Choi et al. have discovered TE expansion in *Fasciola*, which also explained the large lineage-specific genome size and longer annotated gene [6]. We confirmed the result in *F. gigantica* genome and further identified signatures of selection based on unbalanced distribution of inserted TEs between intronic and intergenic regions in history. Especially, the strongest selection signal occurred in the speciation between the *Fasciola* and Fascioloides—a habitat switch from the small intestine to the liver in the host—during the PETM, which accompanied by LINE expansion biased toward intronic regions (Fig 5). This unexpected event provided a new evidence of adaptive evolution driven by transposition events and will prompt investigations of how such differences contribute mechanistically to the morphological phenotypes of liver flukes and related species. There are also many studies in other species supporting the hypothesis that TE invasions endured by organisms have

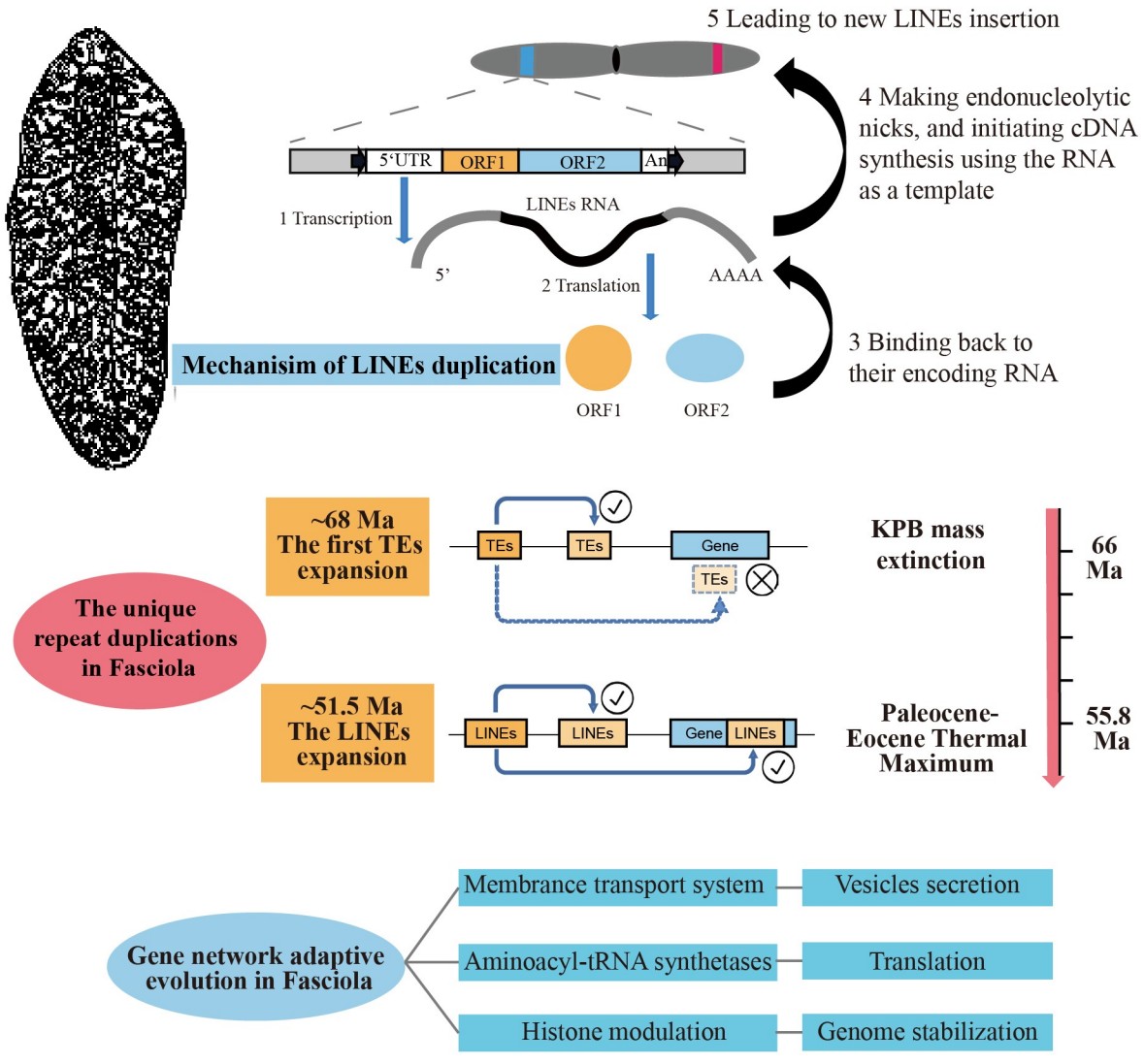

**Fig 5. Schematic diagram of the process of *Fasciola*-specific repeat expansion during evolution.**

catalyzed the evolution of gene-regulatory networks [54]. For example, Eutherian-specific TEs have the epigenetic signatures of enhancers, insulators, and repressors, and bind directly to transcription factors that are essential for pregnancy and coordinately regulate gene expression [55]. Similarly, genes with large-scale insertion of TEs in *Fasciola* species identified here, represent a signature of *Fasciola*-specific evolutionary gene network to distinguish other flukes of the family Fasciolidae. These genes overlap significantly with host-parasite interaction genes, including proteases and E/S proteins, and are enriched in the pathways of EV biogenesis and vesicle trafficking.

The data from genomic, transcriptomic, and proteomic studies can form a good complementary relationship to further our understanding of helminth parasites and their interaction with their hosts. Previous studies have identified a rich source of stage-specific molecules of interest using transcriptomic and proteomic analysis [56,57]. Here, we provided a comprehensive list of predicted E/S proteins in *F. gigantica* and predicted 3300 PPIs at the host-parasite interface, extending our understanding of how the phase I and phase II detoxification enzymes

counteract the effect of ROS. The ability of *Fasciola* species to infect and survive in different tissue environments is underpinned by several key E/S protein gene duplications. Both *Fasciola* species have a common expansion in the secretion of papain-like cysteine peptidase family (Clan A, family C1) [5]. Besides, *F. gigantica* has a specific variation in the *SOD* gene copy number, allowing it to regulate the catalytic activity of the superoxide radical released by the host. The effect of specific gene duplications can also be reflect in the increased body size of *F. gigantica*, which is an important morphometric character to distinguish *Fasciola* species and has a decisive influence on the final host species [58], although a gene level study of this phenotype is barely reported.

Overall, our study demonstrated that the combination of long-read sequencing with Hi-C scaffolding produced a very high-quality liver fluke genome assembly and gene annotation. Additionally, identification of the repeat distribution among the gene regions extended our understanding of the evolutionary process in *Fasciola* species. Further detailed functional studies of secretion might be of great scientific significance to explore their potential application in fascioliasis treatment.

# Materials and methods

## Ethics statement

This study was approved by the Research Ethics Committee of the Guangxi University (Permit code: GXU2019-029). In present study, experiment was performed by the Principle Guidance for the Use and Care of Laboratory Animals.

## Sample collection and *de novo* sequencing

All animal work was approved by the Guangxi University Institutional Animal Care and Use Committee. For the reference genome sequencing, one *F. gigantica* at adult stage was derived from infected buffalo in the Guangxi Zhuang Autonomous Region. Nucleic acids were extracted using a QIAGEN DNeasy (DNA) kit (Qiagen Hilden, Germany). Three *de novo* genome sequencing methods were performed on the liver fluke: We generated (1) 122.4 Gb (~88× depth) PacBio Sequel II single-molecule long reads, with an average read length of 15.8 kb (PacBio, Menlo Park, CA, USA); (2) 89.5 Gb (~66× depth) Illumina HiSeq PE150 pair-end sequencing to correct errors (Illumina, San Diego, CA, USA); and (3) 134 Gb (~100× depth) chromosome conformation capture sequencing (Hi-C) data (sequenced by Illumina platform).

## *De novo* assembly and assessment of the genome quality

A PacBio-only assembly was performed using Canu v2.0 [59,60] using new overlapping and assembly algorithms, including an adaptive overlapping strategy based on *tf-idf* weighted Min-Hash and a sparse assembly graph construction that avoids collapsing diverged repeats and haplotypes. To remove haplotigs and contig overlaps in the assembly, we used Purge_Dups based on the read depth [61]. Arrow (https://github.com/PacificBiosciences/GenomicConsensus) was initially used to reduce the assembly error in the draft assembly, with an improved consensus model based on a more straightforward hidden Markov model approach. Pilon [62] was used to improve the local base accuracy of the contigs via analysis of the read alignment information based on paired-end bam files (thrice). As a result, the initial assembly resulted had an N50 size of 4.89 Mb for the *F. gigantica* reference genome. ALLHiC was capable of building chromosomal-scale scaffolds for the initial genome using Hi-C paired-end reads containing putative restriction enzyme site information (S1 Text) [63]. The whole

genome assembly (contig version) have been deposited in the Genome Warehouse in BIG Data Center under accession number GWHAZTT00000000 and NCBI under Bioproject PRJNA691688.

Three methods were used to evaluate the quality of the genomes. First, we used QUality ASsessment Tool (QUAST) [64] to align the Illumina and PacBio raw reads to the *F. gigantica* reference genome to estimate the coverage and mapping rate. Second, all the Illumina paired-end reads were mapped to the final genome using BWA [65], and single nucleotide polymorphisms (SNPs) were called using Samtools and Bcftools. The predicted error rate was calculated by the homozygous substitutions divided by length of the whole genome, which included the discrepancy between assembly and sequencing data. Thirdly, we assessed the completeness of the genome assemblies and annotated the genes using BUSCO [18].

## Genome annotation

Three gene prediction methods, based on *de novo* prediction, homologous genes, and transcriptomes, were integrated to annotate protein-coding genes. RNA-seq data of *F. gigantica* were obtained from the NCBI Sequence Read Archive, SRR4449208 [66]. RNA-seq reads were aligned to the genome assembly using HISAT2 (v2.2.0) [67] and subsequently assembled using StringTie (v2.1.3) [68]. PASA (v2.4) [69] was another tool used to assemble RNA-seq reads and further generated gene models to train *de novo* programs. Two *de novo* programs, including Augustus (v3.0.2) [70] and SNAP (v2006-07-28) [71], were used to predict genes in the repeat-masked genome sequences. For homology-based prediction, protein sequences from UniRef100 [72] (plagiorchiida-specific, n = 75,612) were aligned on the genome sequence using TBLASTn [73] (e-value $< 10^{-4}$), and GeneWise (version 2.4.1) [74] was used to identify accurate gene structures. All predicted genes from the three approaches were combined using MAKER (v3.1.2) [75] to generate high-confidence gene sets. To obtain gene function annotations, Interproscan (v5.45) [76] was used to identify annotated genes features, including protein families, domains, functional sites, and GO terms from the InterPro database. SwissProt and TrEMBL protein databases were also searched using BLASTp [77] (e-value $< 10^{-4}$). The best BLASTp hits were used to assign homology-based gene functions. BlastKOALA [78] was used to search the KEGG ORTHOLOGY (KO) database. The subsequent enrichment analysis was performed using clusterProfiler using total annotated genes as the background with the "enricher" function [79].

## Repeat annotation and analysis

We combined *de novo* and homology approaches to identify repetitive sequences in our assembly and previous published assemblies, including *F. gigantica*, *F. hepatica*, and *Fasciolopsis buski*. RepeatModeler (v2.0.1) [24] was first used to construct the *de novo* identification and accurate compilation of sequence models representing all of the unique TE families dispersed in the genome. Then, RepeatMasker (v4.1.0) [25] was run on the genome using the combination of *de novo* libraries and a library of known repeats (Repbase-20181026). The relative position between a repeat and a gene was identified using bedtools [80], and the type of repeat was further divided to intronic and intergenic origin. The repeat landscape was constructed using sequence alignments and the complete annotations output from RepeatMasker, depicting the Kimura divergence (Kimura genetic distances between identified repeat sequences and their consensus) distribution of all repeats types. The most notable peak in the repeat landscapes was considered as the most convincing time of repeat duplication in that period. We inferred the time of LINEs insertion by transferring Kimura divergence in RepeatMasker to age (t = d/2mu). The distributions of TE elements were calculated with sliding windows (n = 50). In each

sliding window, we calculated the relative proportion of TE between intronic and intergenic regions, and further corrected them using the whole ratio between intronic and intergenic regions. To calculate mutation rate, we used 559 single-copy orthologs multiple sequence alignment among 8 species produced in the latter gene family analysis, and estimated the mutation rate using MCMCtree with global clock. A Markov chain Monte Carlo (MCMC) process was run for 2,000,000 iterations, with sample frequency of 100 after a burn-in of 1,000 iterations. The median of simulated data was selected as mutation rate (mu = $1.73 \times 10^{-9}$ per base per year).

## Genome-wide host-parasite protein interaction analysis

In addition to the genome data that we generated for *F. gigantica*, we downloaded genome annotation information for human (GCA_000001405.28), swamp buffalo (GWHAAJZ00000000), *F. hepatica* (GCA_002763495.2), *Fasciolopsis buski* (GCA_008360955.1), *Clonorchis sinensis* (GCA_003604175.1), *Schistosoma mansoni* (GCA_000237925.2), and *Taenia multiceps* (GCA_001923025.3) from the NCBI database and BIG Sub (China National Center for Bioinformation, Beijing, China). Proteases and protease inhibitors were identified and classified into families using BLASTp (e-value $< 10^{-4}$) against the MEROPS peptidase database (merops_scan.lib; (European Bioinformatics Institute (EMBL-EBI), Cambridge, UK)), with amino acids at least 80% coverage matched for database proteins. These proteases were divided into five major classes (aspartic, cysteine, metallo, serine, and threonine proteases). E/S proteins (i.e., the secretome) were predicted by the programs SignalP 5.0 [81], TargetP [82], and TMHMM [83]. Proteins with a signal peptide sequence but without a transmembrane region were identified as secretome proteins, excluding the mitochondrial sequences. Genome-wide host-parasite protein interaction analysis was perform by constructing the PPIs between the *F. gigantica* secretome and human proteins expressed in the tissues related to the liver fluke life cycle. For the hosts, we selected human proteins expressed in the small intestine and liver, and located in the plasma membrane and extracellular region. The gene expression and subcellular location information were obtained from the TISSUES [84] and Uniprot (EMBL-EBI) databases, respectively. For *F. gigantica*, secretome molecules were mapped to the human proteome as the reference, using the reciprocal best-hit BLAST method. These two gene datasets were used to construct host-parasite PPI networks. We downloaded the interaction files (protein.links.v11.0) in the STRING database [85], and only highly credible PPIs were retained by excluding PPIs with confidence scores below 0.7. The final STRING network was plotted using Cytoscape [86].

## Gene family analysis

We chose the longest transcript in the downloaded annotation dataset to represent each gene, and removed genes with open reading frames shorter than 150 bp. Gene family clustering was then performed using OrthoFinder (v 2.3.12) [87], based on the predicted gene set for eight genomes, including *F. gigantica* (our assembly), *F. hepatica* (NCBI: GCA_002763495.2), *Fasciolopsis buski* (NCBI: GCA_008360955.1), *Clonorchis sinensis* (NCBI: GCA_003604175.1), *Schistosoma mansoni* (NCBI: GCF_000237925.1), *Taenia multiceps* (NCBI: GCA_001923025.3), swamp buffalo (BIG sub: GWHAAJZ00000000), and human (NCBI: GCF_000001405.39). This analysis yielded 17,992 gene families. To identify gene families that had undergone expansion or contraction, we applied the CAFE (v5.0.0) program [88], which inferred the rate and direction of changes in gene family size over a given phylogeny. Among the eight species, 559 single-copy orthologs were aligned using MUSCLE (v3.8.1551) [89], and we eliminated poorly aligned positions and divergent regions of the alignment using Gblock

0.91b [90]. RAxML (v 8.2.12) was then used with the PROTGAMMALGF model to estimate a maximum likelihood tree. Divergence times were estimated using PAML MCMCTREE [91]. A Markov chain Monte Carlo (MCMC) process was run for 2,000,000 iterations, with a sample frequency of 100 after a burn-in of 1,000 iterations under an independent rates model. Two independent runs were performed to check the convergence. The fossil-calibrated eukaryote phylogeny was used to set the root height for the species tree, taken from the age of Animals (602–661 Ma) estimated in a previous fossil-calibrated eukaryotic phylogeny [92] and the divergence time between the euarchontoglires and laurasiatheria: (95.3–113 Ma) [93].

To enhance the reproducibility of the results, we deposit the laboratory protocols in protocols.io (PROTOCOL DOI): http://dx.doi.org/10.17504/protocols.io.bxatpien.

## Supporting information

**S1 Fig. Genome-wide all-by-all chromosome conformation capture sequencing (Hi-C) interaction in *F. gigantica* (Bins = 500 K).**
(TIF)

**S2 Fig. Comparison of chromosome length between the chromosome conformation capture sequencing (Hi-C) assembly and estimates from published karyotype data by Jae Ku Rhee.**
(TIF)

**S3 Fig. Boxplot of average gene length.**
(TIF)

**S4 Fig. Boxplot of average coding sequence (CDS) length per gene.**
(TIF)

**S5 Fig. Divergence distribution of classified families of transposable elements.** The classified transposon families in *F. gigantica*.
(TIF)

**S6 Fig. Expansion time of long terminal repeats (LTRs) and long interspersed elements (LINEs).** The mutation rate was $1.73 \times 10^{-9}$.
(TIF)

**S7 Fig. Estimation of *F. gigantica* genome size based on the expansion time of repeat sequences during evolution.** The mutation rate was $1.73 \times 10^{-9}$.
(TIF)

**S8 Fig. Overlapping E/S proteins between this study and proteomic study by Di Maggio LS et al [94].**
(TIF)

**S1 Table. Genome sequencing strategy for buffaloes.**
(XLSX)

**S2 Table. Summary of the *Fasciola gigantica* genome assembly.**
(XLSX)

**S3 Table. Summary of different assemblies in *Fasciola* species.**
(XLSX)

**S4 Table. Summary of chromosome conformation capture sequencing (Hi-C) assembly of the chromosome length in *Fasciola gigantica*.**
(XLSX)

**S5 Table. Assessment of the completeness and accuracy of the genome.**
(XLSX)

**S6 Table. BUSCO assessment of the genome.**
(XLSX)

**S7 Table. Number of genes with functional classification gained using various methods.**
(XLSX)

**S8 Table. Transposable element content of *Fasciola gigantica* genome.**
(XLSX)

**S9 Table. The list of genes with more than 10 kb of long interspersed element (LINE) insertion between 41 Ma and 62 Ma.**
(XLSX)

**S10 Table. Gene ontology (GO) term category enrichment for genes with more than 10 kb of long interspersed element (LINE) insertion between 41 Ma and 62 Ma.**
(XLSX)

**S11 Table. Kyoto Encyclopedia of Genes and Genomes (KEGG pathway enrichment for genes with more than 10 kb of long interspersed element (LINE) insertion between 41 Ma and 62 Ma.**
(XLSX)

**S12 Table. Kyoto Encyclopedia of Genes and Genomes (KEGG) pathway enrichment for genes with more than 10 kb of long interspersed element (LINE) insertion between 41 Ma and 62 Ma.**
(XLSX)

**S13 Table. Protein inhibitors in the *Fasciola gigantica* genome.**
(XLSX)

**S14 Table. Excretory/secretory (E/S) proteins in the *Fasciola gigantica* genome.**
(XLSX)

**S15 Table. Gene ontology (GO) term category enrichment for excretory/secretory (E/S) proteins.**
(XLSX)

**S16 Table. Gene ontology (GO) term category enrichment for rapidly evolving families specific to *F. gigantica*.**
(XLSX)

**S17 Table. Function annotation based on human uniprot gene using blastp with E-value < 10−4.**
(XLSX)

**S1 Text. AGP file for *Fasciola gigantica*.txt.**
(DOC)

## Author Contributions

**Conceptualization:** Kuiqing Cui, Zhiqiang Wang, Zhipeng Li.

**Data curation:** Kuiqing Cui, Zhiqiang Wang, Zhipeng Li.

**Formal analysis:** Kuiqing Cui, Zhiqiang Wang, Zhipeng Li.

**Funding acquisition:** Qingyou Liu.

**Investigation:** Xier Luo.

**Methodology:** Xier Luo, Jue Ruan.

**Project administration:** Xier Luo, Qingyou Liu.

**Resources:** Zhipeng Li, Zhengjiao Wu, Weiyi Huang.

**Software:** Xier Luo, Jue Ruan.

**Supervision:** Jue Ruan, Qingyou Liu.

**Validation:** Zhengjiao Wu, Weiyu Zhang.

**Visualization:** Weiyu Zhang, Qingyou Liu.

**Writing – original draft:** Xier Luo.

**Writing – review & editing:** Weiyi Huang, Xing-Quan Zhu, Jue Ruan, Weiyu Zhang, Qingyou Liu.

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
