## [Decision Letter · Decision Letter 0]

24 May 2021

Dear Dr. Liu,

Thank you very much for submitting your manuscript "High-quality reference genome of Fasciola gigantica: Insights into the genomic signatures of transposon-mediated evolution and specific parasitic adaption in tropical regions" for consideration at PLOS Neglected Tropical Diseases. As with all papers reviewed by the journal, your manuscript was reviewed by members of the editorial board and by several independent reviewers. In light of the reviews (below this email), we would like to invite the resubmission of a significantly-revised version that takes into account the reviewers' comments. 

We cannot make any decision about publication until we have seen the revised manuscript and your response to the reviewers' comments. Your revised manuscript is also likely to be sent to reviewers for further evaluation.

Sincerely,

Neil David Young

Associate Editor

Makedonka Mitreva

Deputy Editor

Reviewer's Responses to Questions

**Key Review Criteria Required for Acceptance?**

**Methods**

-Are the objectives of the study clearly articulated with a clear testable hypothesis stated?

-Is the study design appropriate to address the stated objectives?

-Is the population clearly described and appropriate for the hypothesis being tested?

-Is the sample size sufficient to ensure adequate power to address the hypothesis being tested?

-Were correct statistical analysis used to support conclusions?

-Are there concerns about ethical or regulatory requirements being met?

Reviewer #1: 1. How many parasites were used for the DNA extraction and from which life cycle stage? If more than one parasite was used detail how the sequencing libraries were prepared and from which samples.

2. Lines 454-456 - detail genome/transcriptome assemblies used here, provide accession/identifier numbers.

3. Line 502 - include details of the 8 genomes used here - species names and genome identifiers.

Reviewer #2: Major points:

1. Lines 469-470: The Authors should clarify which species were uses to produce single copy orthologs and where the underlying evidence to estimate the speciation events among these species was collected from? Did the author have for instance estimates from fossil data for calibration? Also, the authors should give more detailed description for the calculation of mutation rate because plain referral to MCMC and CDS alignments do not make the method repeatable for other researches.

Reviewer #3: The idea behind the study is clear and sound.

Methods in genome sequencing, assembly and annotation seem appropriate. 

Some of the downstream analysis would need a bit of improvement

**Results**

-Does the analysis presented match the analysis plan?

-Are the results clearly and completely presented?

-Are the figures (Tables, Images) of sufficient quality for clarity?

Reviewer #1: 1. Lines 42 and in the results section - the authors describe how they identified ES proteins that were used to investigate protein-protein interactions between host and parasite. 

a. How were these proteins identified/classified? 

b. Did the authors base their predictions solely on whether the proteins contained a signal peptide and therefore would be likely to be secreted? 

c. Analysis from F. hepatica ES proteome studies has shown that proteins that do not have signal peptides are also present in the ES products, particularly within the EVs. Did the authors check their predicted ES proteins to known ES proteome datasets for Fasciola to confirm this list? 

d. The authors should mention that these proteins represented the predicted secretome/ES proteins rather than stating these are ES proteins, unless confirmed. This data also impacts on how the PPI data should be interpreted. 

2. Figures - all the figures are composite figures - the authors should modify the font size of the text so that it will be visible upon publication.

Reviewer #3: The article provide a significant improvement in the knowledge of the genome of F.gigantica, that might be a very useful resource.

In general the results are presented in a clear way, although some clarifications and improvements can be made as detailed below.

In general the figures are of good quality, the supplementary information provided in tables need improvement, particularly by providing a reasonable annotation of individual recognizable genes.

**Conclusions**

-Are the conclusions supported by the data presented?

-Are the limitations of analysis clearly described?

-Do the authors discuss how these data can be helpful to advance our understanding of the topic under study?

-Is public health relevance addressed?

Reviewer #1: (No Response)

Reviewer #3: Most of the ideas behind the article are sound and supported by the data, although some analysis might need to be revised. 

The discussion in general stress relevant points, that are open for discussion

The subject is relevant and the data generated might be a valuable resource

**Editorial and Data Presentation Modifications?**

Reviewer #1: 1. Line 28 and throughout the manuscript – fascioliasis should not be in italics, however, Fasciola should be in italics.

2. Line 59 – prevalence of F. gigantica infection.

3. Line 208 – sentence should read as ‘genes have’ or ‘gene has’ depending on the number of genes the authors are referring to.

4. Line 333 - of the rapidly evolving

5. Line 364 - networks

6. Lines 226-227 - rationale for abbreviation AAASs when the abbreviation for aminoacyl tRNA synthetases is typically AARSs- also why is the abbreviation in italics? The AARSs abbreviation is used on line 227 but is not defined.

7. References:

a. Lines 85-86 – Reference [1] is not the correct reference for this statement relating to the mode of action of TCBZ in disrupting beta tubulin polymerization.

b. References 7 and 45 are the same; References 10 and 29 are the same.

Reviewer #2: Minor points

1. Line 165: would -> would be

2. Line 281: is mainly -> mainly

3. Line 304: emphasize -> emphasizes

4. Line 469: 173x10-9 -> 173x10-9 per basepair per year

Reviewer #3: A minor suggestion, I feel that swapping some paragraphs in the intro might improve the understanding without changing the meaning. 

Fisrt sentence of last paragraph (lines 92-93) would be better after line 62, that can be followed by the 3rd paragraph of the intro (lines 77 -91). This would join all the info on the biology of the parasite, before presenting the status on trematode omics and particularly Fasciolidae data (lines 63-76).

**Summary and General Comments**

Reviewer #1: The manuscript by Luo and colleagues describes a high quality genome assembly for Fasciola gigantica. This genome dataset represents an improved dataset for the tropical liver fluke, which has been assembled into 10 potential chromosomes. This study paves the way for improvements for the other Fasciola species genome allowing future comparatative analyses and provides novel insight for liver fluke biology. The manuscript is well written and is suitable for publication in PLoS NTD.

Reviewer #2: This manuscript introduces reference genome for a socio-economically important trematode Fasciola gigantica and hypothesizes both the role of both transposon expansion event and uses single copy gene to infer speciation events between Fasciola and Fasciolopsis and between Fasciola gigantica and Fasciola hepatica, respectively. Moreover, the authors hypothesize the role of gene family expansions to the size of the trematode and its development to epidemic in tropical and subtropical regions.

The assembled genome for F. gigantiga described in this manuscript clearly has major importance to the community studying parasites and the diseases they cause. The manuscript is well written, the methods for the assembly and annotation follow currently known best practices (I could only slightly criticize the use of SNAP in de novo prediction).

Apart from the required clarification to the calculation of the mutation rate, and few minor enhancements, there are not objections to publish the manuscript.

Reviewer #3: The manuscript presented by Luo et al provides a novel assembly of the genome of the liver fluke Fasciola gigantica, and analyze the plausible role of transposons in driving adaptation in this species.

The work is well written and contributes a good wealth of data, providing an almost chromosome level assembly of this relevant species, with a good deal of info on supplementary data, that would undoubtedly move forward the knowledge of Fasciolidae biology. I feel the article has a lot of strengths, but also some weakness that might be improved a bit in order to get it published. These lay essentially in the analysis of the repeats and the prediction of interactions between parasite and host genes, and the absence of an annotated list of genes that should be provided. 

Anyway I provide below a more detailed account of doubts and questions that I feel can clarify the findings and their relevance.

1. The first section describes the novel assembly that integrates long reads, paired reads and HiC data. This results in a slightly larger genome than previously published (refs 5-8), consistent with a better resolution of probably collapsed repeat sequences that here are resolved. Gene content is similar to previously described genomes of Fasciolidae, although functional annotation results in roughly 2/3 of genes with assigned function (either on Interpro, GO or KEGG). I wonder from table S7 how many genes have no associated function whatsoever, is a minor point, but would stress the size of the unknown bin. 

2. When looking at all the supplementary data, I noticed that an annotation file with the identified genes and their putative annotation is missing. This is a relevant tool that should be included, either as a supplementary table and/or as a gff annotation file with the genome assembly. 

3. The notion that genes were larger in Fasciolidae than in other trematodes (lines 129-131) have already been advanced, and figures S3 and s4 are almost identical to figure 1C from Choi et al, ref 7, that also includes a comparison of intron lengths. This should be properly referred and discussed. By the way ref 7 and 45 are the same…please correct.

4. The second section focus on the repeats expansions that resulted in larger genomes. The authors estimate two expansions events and time them to 12 and 65 Ma (lines 153-155). I feel that the procedures for the estimation of these times need to be further clarified. 

5. The authors made the reasonable hypothesis that they might be underlying the increase in genome size, and probably providing evolutive advantages (lines 156-160). This idea, and the distribution of repeats between intergenic and intronic regions (next point) were also previously advanced by Choi et al, and the similarities and differences should be acknowledged. 

6. The comparison of the distribution of repeats between intronic and intergenic regions (lines 161-185) is a reasonable thing to do, but I feel that the description of the procedures and results are not fully clear. I assume that “the relative proportion of TEs between intronic or intergenic regions” is calculated in all intronic vs all intergenic repeats. What is the “inferred size” of these regions? What is the corrective repeat ratio that appears in Fig 2B and how is calculated?

7. The idea of purifying selection acting on intronic regions, except for LINEs (line 172) would imply that a statistical significant difference in TE types within intron exists in comparison with intergenic repeats. I feel this is missing, or I cannot figure out this from fig 2B. 

8. While this transmits the idea of LINEs being enriched in intronic sequences, a different picture has been presented previously by Choi et al, showing LINE enrichment both intronic and intergenic (see Fig.2B of ref 7) but possibly consisting of different LINE types. Are the results consistent to this? It would be good to discuss it.

9. The idea of selecting a set of enlarged Fasciola introns and compare to other trematode species is a good one, and searching for functions through GO enrichment also is reasonable (187-197), but I feel that the functions highlighted do not reflect the diversity described in Fig 2C or in the supplementary tables (by the way, they are slightly different, there are missing categories in one or the other). 

10. Although is a good proxy to what might be happening, I found strange that some highly represented categories as signal transduction was not highlighted, especially considering that components such as protein phosphorylation or GTPase activators are also enriched.

11. Several questions come to my mind on the finding of particular groups of genes like tRNA synthetases (lines 225-231), or histone modificators (lines 234-240) as enriched in long introns. For example, are there are other copies of these genes in the genome that were not affected by the inclusion of long introns? Are the large introns in a particular position (more 5’ or 3’) within the genes? Intersting…

12. The fourth section attempts to find genes that interact with those of the host, and here is where I have more concerns. First proteases and inhibitors are identified by comparison with the merops database (lines 244-257), however, the annotation provided does not allow to compare if these results have, for example improved the annotation of these relevant gene families in relation to previously published assemblies (refs 5-8). By analyzing the supplementary tables I noticed that the gene annotation based on this assembly is not provided. The tables provides access to merops entries, but does not correlate with annotated genes in FGIG. I assume that repeated merops entries that appear in the table correspond to diverse genes in the FGIG genome that are best match to these merops entries, but this need to be clarified. Gene names should be corrected providing a unique identifier for each gene as well as their putative annotation. This is particularly relevant in gene families as for example those that appear in suppl table 15. It seems that approximately 17 Legumain genes have been found, but if all share the same name is impossible to evaluate for example if they are differentially expressed in diverse life stages or tissues. A complete annotation of the genes list should be provided as previously stated (in point 2).

13. The major concern is however, the assumption of interactions based on the protein-protein interaction (lines 260-280). I feel this analysis is essentially incorrect. Interactions were inferred by selecting host intestinal and liver surface and secreted proteins, and those also expressed by the parasite. I wonder how many of the FGIG surface proteins mapped to human counterparts. Beside this, the interactions reported seem to be based on those described for the mammalian counterparts of the FGIG proteins. These are proved interactions within mammals, but there is no line of evidence so far that the same interactions would take place with heterologous proteins from the parasite. Extreme care should be taken into inferring biologically significant interactions without further experimental evidence. I feel this whole section is over interpreting the results. 

14. I also have some doubts about the gene family analysis, since some of the F.gigantica specific families highlighted correspond to quite conserved GO functions, that surely are present in F.hepatica and other trematodes. Consequently they should represent duplication events exclusive of F.gigantica, that can be easily analyzed in more detail. Unfortunately the poor annotation of genes does not allow to test this since rather than having a list of genes that constitute the family we have a database hitname repeated as many times as copies. As already mentioned good annotation would allow a better understanding of the results.

In general I feel that the article has interesting novel data, but have some weakness in the analysis, most of them that can be reasonably sorted out before publication.

PLOS authors have the option to publish the peer review history of their article (what does this mean?). If published, this will include your full peer review and any attached files.

Reviewer #1: No

Reviewer #2: No

Reviewer #3: No
---

## [Decision Letter · Decision Letter 1]

28 Jul 2021

Dear Dr. Liu,

Thank you very much for submitting your manuscript "High-quality reference genome of Fasciola gigantica: Insights into the genomic signatures of transposon-mediated evolution and specific parasitic adaption in tropical regions" for consideration at PLOS Neglected Tropical Diseases. As with all papers reviewed by the journal, your manuscript was reviewed by members of the editorial board and by several independent reviewers. The reviewers appreciated the attention to an important topic. Based on the reviews, we are likely to accept this manuscript for publication, providing that you modify the manuscript according to the review recommendations. 

Thank you for submitting your rejoinder. Please submit a word document with changes in the document highlighted or using tracked changes. Please make sure that questions raised by each reviewer are addressed in the manuscript. Also, please include a table with a summary of the annotation linked to each gene. Please see other minor comments from the two reviewers.

Sincerely,

Neil David Young

Associate Editor

Makedonka Mitreva

Deputy Editor

Thank you for submitting your rejoinder. Please submit a word document with changes in the document highlighted or using tracked changes. Please make sure that questions raised by each reviewer are addressed in the manuscript. Also, please include a table with a summary of the annotation linked to each gene. Please see other minor comments from the two reviewers.

Reviewer's Responses to Questions

**Key Review Criteria Required for Acceptance?**

**Methods**

-Are the objectives of the study clearly articulated with a clear testable hypothesis stated?

-Is the study design appropriate to address the stated objectives?

-Is the population clearly described and appropriate for the hypothesis being tested?

-Is the sample size sufficient to ensure adequate power to address the hypothesis being tested?

-Were correct statistical analysis used to support conclusions?

-Are there concerns about ethical or regulatory requirements being met?

Reviewer #3: Some methodological issues still present see below.

**Results**

-Does the analysis presented match the analysis plan?

-Are the results clearly and completely presented?

-Are the figures (Tables, Images) of sufficient quality for clarity?

Reviewer #3: Please see below in general comments

**Conclusions**

-Are the conclusions supported by the data presented?

-Are the limitations of analysis clearly described?

-Do the authors discuss how these data can be helpful to advance our understanding of the topic under study?

-Is public health relevance addressed?

Reviewer #3: Fine. ssee below

**Editorial and Data Presentation Modifications?**

Reviewer #3: Some changes in the way data are presented are still needed. See below

**Summary and General Comments**

Reviewer #1: The authors have addressed the comments raised by the previous review - however several of the points included in their response have not been incorporated into the manuscript (see below). Following the edits/comments below the manuscript will be suitable for publication.

1. For example - the authors stated that only one adult fluke was just for the sequencing. This needs to be included in the methods. Please include where relevant the points raised in the review in the text. 

2. Also it was difficult to see where the changes were made as the authors used the comment box on the pdf to add in their changes. The edits apparently made in the highlighted manuscript, such as lines 161-172 relating to the purifying selection, were not incorporated in the final version. In the next review please include a highlighted manuscript using tracked changes or highlighting the text in colour so that the changes can be followed.

3. Line 105 - The reference included for the beta tubulin polymerization is still not right. Include a relevant reference.

4. Lines 221-224 - Although the authors state that they checked their predicted secreted sequences against the EV proteins published by de la Torre et al., the authors should also check their data against the available secretome data not just the EV proteins, to identify the actual proteins present in the ES products.

5. Line 309-310 - how did the authors perform this analysis? Is this based on their predicted secretome data or from proteomic analysis?

6. Line 383 - amend the sentence to: Choi et al. 

7. Lines 544-547 - The species names in this section should be in italics.

Reviewer #3: COMMENTS ON REVISED VERSION OF LUO ET AL.

While most of the issues and questions have been considered, some of the main issues are still there, and need to be corrected in order to make the manuscript acceptable for publication.

1. The main issue is still the poor annotation, devoid of unique Identifiers for the genes, that make most of the data presented in tables and figures completely unfollowable. While the manuscript is an excellent effort assembling a chromosome level genome, the usefulness of this resource for the community upon publication would be limited by this issue, making it not comparable to other assemblies available. 

Following one of the suggestions a gff table is included (A23). While that provides a detailed information for those more interested, is not a solution for the simple reader interested in looking the information of any one of the genes that appear in tables 9-16. There is no place to go, and since genes are named based on their hits, when these are repeated (as frequently occur) there is no way of identifying a particular gene and differentiate it from other members of the same family. Is impossible to pretend that each reader interested in a particular gene should download and filter a gff table (as suggested in responses A33 and A35). Providing appropriate and clear information is a task of the authors, not the readers. 

2. A very simple solution for this is a table with the 12503 identified genes each with a unique IDs and a definition. Beside this, all tables should include gene IDs (it can also combine names but unique identifiers are mandatory) in order to make it useful and verifiable. This is a main issue and should be solved before publication. The work has reached excellent results assembling the genome, and annotation, and it would be a pity if published in a way that requires other researchers to run complex filtering steps and/or reanalize the data to find information that the authors already have, and should be made available in a way that could be easily used by the community. 

3. The annotation weakness also compromises the idea of the protein-protein interaction analysis. I agree as stated in response A34 that this could be an approach to gain information based on better analyzed protein interactions in host, but we have to be very careful on the design of the assay and on the interpretation of the results. In order to make this kind of analysis, we have to be sure that the proteins considered are true orthologs, ie. that the actual parasite gene considered is the unique orthologue of the host one detected in the interaction. Several of the proteins included might be part of gene families, so the question of finding the correct orthologue (and not paralogues) is not trivial. While is stated that the reciprocal blast best-hits were considered, the absence of proper identifiers make this untestable. Here again, having good unique gene ids is essential.

4. Similarly in relation to the gene family analysis, is not a question of if is reasonable to believe in the results as stated in answer 33 but rather if the information provided sustain the results. I do believe that the authors have performed well the experiments and have sound results, but I cannot find useful information in what is published, since I cannot identify different genes of the same family. Questions related to these families, their origins and similarities with those from other species remain obscure if we cannot pick the appropriate genes to compare and advance in their knowledge. 

5. Other issues related to the repeats and the acknowledge and discussion of previous works in the same line have been taken into account, making minor modification in the discussion (responses A24, A26 and A29). Since some of the results here mirror those already published I would have expected them to be commented earlier than in discussion, but this is a matter of opinion.

6. Other questions related to repeat expansion analysis and selection (A25-28), were answered or clarified appropriately. Thanks for that.

Figure Files:

Data Requirements:

Reproducibility:

References

---

## [Editor Report · Decision Letter 2]

10 Aug 2021

Dear Dr. Liu,

Thank you very much for submitting your manuscript "High-quality reference genome of Fasciola gigantica: Insights into the genomic signatures of transposon-mediated evolution and specific parasitic adaption in tropical regions" for consideration at PLOS Neglected Tropical Diseases. As with all papers reviewed by the journal, your manuscript was reviewed by members of the editorial board and by several independent reviewers. The reviewers appreciated the attention to an important topic. Based on the reviews, we are likely to accept this manuscript for publication, providing that you modify the manuscript according to the review recommendations. 

Thank you for addressing comments from each reviewer. I am satisfied that you have addressed their comments and improved the quality of the manuscript.

Please format the references correctly and resubmit the manuscript.

Please check format suggested for PLoS NTD and then consider the specific modifications:

The species name should be italicised

Decision should be made on the use of sentence case or capitalise every word in title

Ensure consistency of information displayed (e.g. doi numbers, PubMed IDs)

Sincerely,

Neil David Young

Associate Editor

Makedonka Mitreva

Deputy Editor

Thank you for addressing comments from each reviewer. I am satisfied that you have addressed their comments and improved the quality of the manuscript.

Please format the references correctly and resubmit the manuscript.

Please check format suggested for PLoS NTD and then consider the specific modifications:

The species name should be italicised

Decision should be made on the use of sentence case or capitalise every word

Ensure consistency of information displayed (e.g. doi numbers, PubMed IDs)

Figure Files:

Data Requirements:

Reproducibility:

References

---

## [Editor Report · Decision Letter 3]

18 Aug 2021

Dear Dr. Liu,

Thank you very much for submitting your manuscript "High-quality reference genome of Fasciola gigantica: Insights into the genomic signatures of transposon-mediated evolution and specific parasitic adaption in tropical regions" for consideration at PLOS Neglected Tropical Diseases. As with all papers reviewed by the journal, your manuscript was reviewed by members of the editorial board and by several independent reviewers. The reviewers appreciated the attention to an important topic. Based on the reviews, we are likely to accept this manuscript for publication, providing that you modify the manuscript according to the review recommendations. 

In the R2 version of this manuscript, i can see references that are still incorrectly formatted. 

There are several discrepancies with the format of your reference list. 

Reference format needs to be unified and the following addressed:

Check format suggested for PLoS NTD and then consider the specific modifications:

The species names must be italicised

Decision should be made on the use of sentence case or to capitalise every word or letter. There is a mix of all styles at the moment. 

Ensure consistency of information displayed (e.g. doi numbers, PubMed IDs).

Sincerely,

Neil David Young

Associate Editor

Makedonka Mitreva

Deputy Editor

In the R2 version of this manuscript, i can see references that are still incorrectly formatted. 

There are several discrepancies with the format of your reference list. 

Reference format needs to be unified and the following addressed:

Check format suggested for PLoS NTD and then consider the specific modifications:

The species names must be italicised

Decision should be made on the use of sentence case or to capitalise every word or letter. There is a mix of all styles at the moment. 

Ensure consistency of information displayed (e.g. doi numbers, PubMed IDs).

Figure Files:

Data Requirements:

Reproducibility:

References

---

## [Editor Report · Decision Letter 4]

23 Aug 2021

Dear Dr. Liu,

We are pleased to inform you that your manuscript 'High-quality reference genome of Fasciola gigantica: Insights into the genomic signatures of transposon-mediated evolution and specific parasitic adaption in tropical regions' has been provisionally accepted for publication in PLOS Neglected Tropical Diseases.

Best regards,

Neil David Young

Associate Editor

Makedonka Mitreva

Deputy Editor

---

## [Editor Report · Acceptance letter]

1 Oct 2021

Dear Dr. Liu,

We are delighted to inform you that your manuscript, "High-quality reference genome of Fasciola gigantica: Insights into the genomic signatures of transposon-mediated evolution and specific parasitic adaption in tropical regions," has been formally accepted for publication in PLOS Neglected Tropical Diseases.

Best regards,

Shaden Kamhawi

co-Editor-in-Chief

Paul Brindley

co-Editor-in-Chief
